# Total syntheses of Tetrodotoxin and 9-*epi*Tetrodotoxin

Peihao Chen[1,2,6], Jing Wang [2,3,4,6], Shuangfeng Zhang[2], Yan Wang[2,3,4], Yuze Sun[2], Songlin Bai[2,3,4], Qingcui Wu[2], Xinyu Cheng[2,5], Peng Cao [2,4] & Xiangbing Qi [2,4] ✉

Tetrodotoxin and congeners are specific voltage-gated sodium channel blockers that exhibit remarkable anesthetic and analgesic effects. Here, we present a scalable asymmetric syntheses of Tetrodotoxin and 9-*epi*Tetrodotoxin from the abundant chemical feedstock furfuryl alcohol. The optically pure cyclohexane skeleton is assembled via a stereoselective Diels-Alder reaction. The dense heteroatom substituents are established sequentially by a series of functional group interconversions on highly oxygenated cyclohexane frameworks, including a chemoselective cyclic anhydride opening, and a decarboxylative hydroxylation. An innovative $SmI_2$-mediated concurrent fragmentation, an oxo-bridge ring opening and ester reduction followed by an Upjohn dihydroxylation deliver the highly oxidized skeleton. Ruthenium-catalyzed oxidative alkyne cleavage and formation of the hemiaminal and orthoester under acidic conditions enable the rapid assembly of Tetrodotoxin, anhydro-Tetrodotoxin, 9-*epi*Tetrodotoxin, and 9-*epi* lactone-Tetrodotoxin.

Tetrodotoxin (TTX, 1) is one of the most potent neurotoxins with a complex structure and analgesic effects. After the first isolation of TTX in 1909[1], the structure of this highly polar zwitterion was solved by Woodward[2,3], Tsuda[4], Goto[5], and Mosher[6] simultaneously in 1964 using degradative methods and NMR spectroscopy. TTX's unique structure comprises a densely heteroatom-substituted, stereochemically complex framework that has a rigid dioxa-adamantane cage with an ortho acid, a cyclic guanidinium hemiaminal moiety, and nine contiguous stereogenic centers, including one bridgehead nitrogen-containing quaternary center. There are three compounds in equilibrium—ortho ester, 4,9-anhydro, and lactone, that are known to interconvert under acidic conditions[7,8]. Recently, the TTX analogue 9-*epi*Tetrodotoxin (1a) was isolated as an equilibrium mixture of the hemilactal and 10,8-lactone forms[9]. TTX is neurotoxic and exhibits prominent anesthetic and analgesic properties in animal models. The mode of action of this bipolar molecule is defined by its disruption of voltage-gated sodium ion channels ($Na_v$), which was originally suggested in the early 1960s[10,11] and was recently

confirmed by crystallographic studies[12,13]. Extensive pharmacological investigations, including clinical trials[14,15], have demonstrated the potential promise of TTX in pain treatment and detoxification from heroin addiction[16]; accordingly, a reliable source of TTX is of practical significance.

The remarkably polar functionality, stereochemically complex architecture, and fascinating biological activity have made this compound an attractive synthetic target. To date, numerous efforts have been made towards the total synthesis of TTX (Fig. 1a), with the first synthesis by Kishi in 1972 (Fig. 1b)[17,18]. Subsequently, asymmetric syntheses have been achieved by Isobe[8,19], Du Bois[20], Sato[21–23], Fukuyama[24], Yokoshima[25], and Marin[26]. More recently, Trauner described an elegant and concise asymmetric synthesis of TTX based on a glucose derivative[27]. In addition to these syntheses, TTX has also been a model compound for demonstrating creative synthetic strategies in assembling this type of highly oxygenated guanidinium alkaloids efficiently (Keana[28–30], Burgey[31], Alonso[32–35], Taber[36], Shinada[37], Ciufolini[38,39], Hudlicky[40], Nishikawa[41–44] and Johnson[45,46]).

[1]School of Life Sciences, Peking University, Beijing 100871, China. [2]National Institute of Biological Sciences, 7 Science Park Road, Zhongguancun Life Science Park, Beijing 102206, China. [3]School of Life Sciences, Tsinghua University, Beijing 100084, China. [4]Tsinghua Institute of Multidisciplinary Biomedical Research, Tsinghua University, Beijing 100084, China. [5]National Institute of Biological Sciences, Chinese Academy of Medical Sciences&Peking Union Medical College, Beijing 100730, China. [6]These authors contributed equally: Peihao Chen, Jing Wang. ✉e-mail: qixiangbing@nibs.ac.cn

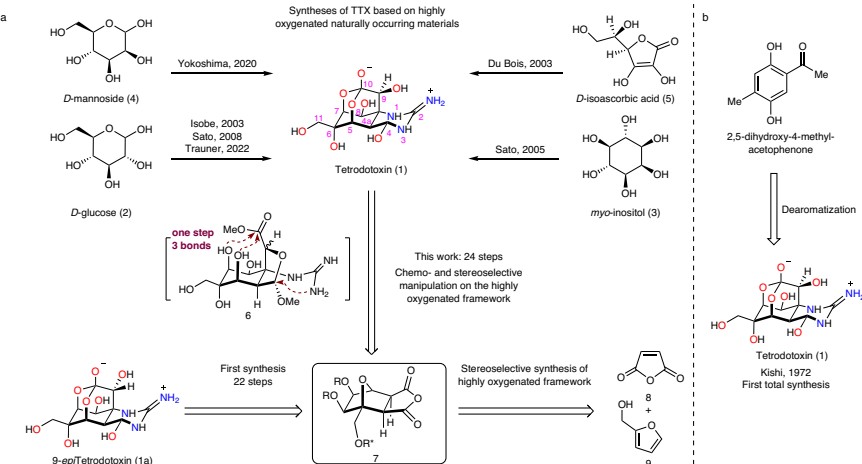

**Fig. 1 | Previous syntheses of Tetrodotoxin and retrosynthetic analysis.**
**a** Syntheses of TTX based on naturally occurring materials and retrosynthetic analysis of TTX. **b** Kishi's first total synthesis of TTX. The oxygen atoms of TTX and
9-*epi*TTX are highlighted in red, and the nitrogen atoms are highlighted in blue. The TTX number is shown in pink.

Precise functional group manipulations on heavily heteroatom-substituted, stereochemically complex frameworks have proven challenging, as evidenced by the total synthesis of highly oxidized natural products[47–53], and as exemplified by the synthetic studies of TTX by Isobe[8], Du Bios[20], Sato[21,22], Yokoshima[25], and Trauner[27] (Fig. 1a) using highly oxygenated natural starting materials such as D-glucose (2), myo-inositol (3), D-mannoside (4), or D-isoascorbic acid (5). Although the preexisting oxygen functionality in these naturally occurring materials provides the functionality basis of TTX, efficient and precise interconversion of these similar functionalities on the densely heteroatom-substituted skeleton in a chemo and stereoselective manner is arduous. We envisioned that if the highly oxygenated framework could be assembled rapidly in the early stage in a stereo-controllable fashion and followed by sequential chemo and stereoselective functional group manipulations might provide a practical solution to a concise synthesis of TTX and its congeners (Fig. 1a). Here; we describe a distinct synthetic strategy that streamlines the incorporation of the dense heteroatom-substituted architecture and is amenable to a scalable synthesis of 9-*epi*TTX and TTX (>15 mg, which is the largest scale known in literature).

## Results

### Retrosynthetic analysis
Retrosynthetic analysis (Fig. 1a) reveals that the hemiaminal and orthoester moieties of the complex dioxa-adamantane architecture can be obtained in one step from intermediate 6, in which both the ester and guanidinium groups are built upon the bridgehead oxygen functionality of framework 7 via a series of well-planned events: SmI₂-mediated reductive oxo-bridge ring opening, Dess–Martin oxidation, chloroepoxidation[22,54,55], stereoselective epoxide opening, and ruthenium-catalyzed oxidative alkyne cleavage. The anhydride motif of 7 is initially transformed into chemically differentiated mono-acid and mono-ester by regioselective methanolysis, which lays the foundation for subsequent radical decarboxylative hydroxylation and hemiaminal synthesis from the redox manipulation of the ester. To access the highly oxygenated chiral framework 7, a stereoselective strategy is proposed from a chiral auxiliary assisted Diels–Alder reaction of the easily accessible maleic anhydride 8 and furfuryl alcohol 9.

### Total syntheses of Tetrodotoxin and 9-*epi*Tetrodotoxin
The synthesis of TTX (1) was initiated with the stereoselective construction of the oxygen-substituted cyclohexane skeleton (Fig. 2). The first oxygen functionality was derived from furfuryl alcohol 9 directly.

Esterification of furfuryl alcohol 9 with chiral auxiliary (−)-(1S)-camphanic acid afforded ester 10 (Supplementary Fig. 1). To achieve the enantiomerically pure 7-oxabicyclo[2.2.1]hept-2-ene derivative 11[56], we developed a reliable stereoselective Diels–Alder protocol by heating 10 with maleic anhydride in the presence of isopropyl ether as the solvent (Fig. 3a and Supplementary Tab.1). Initially, the original protocol by Vogel[56] under neat conditions was attempted, but only a 5:4 mixture of two inseparable *exo* adducts 11 and 11a was observed (by ¹H-NMR analysis of the reaction mixture). Investigation of reaction conditions, including the effects of the molar ratio of reactants, Lewis acids, reaction time, and temperature, was unfruitful in terms of either yield or diastereoselectivity.

Consequently, a survey of solvents was conducted, and the use of isopropyl ether was found to be the crucial factor for the successful generation of optically pure diastereomer 11 as a single detectable *exo* cycloadduct (ratio of 11: 11a > 20:1). The high *exo*-selectivity observed in the current cycloaddition presumably results from the retro-Diels–Alder fragmentation of unstable *endo* cycloadduct[57]. However, whether the chiral auxiliary (−)-(1S)-camphanic acid plays a stereoselective control for Diels–Alder cycloaddition or promotes the crystallization-based enrichment is still a puzzle since no other diastereomers were detected during the whole process, which is inconsistent with the observation by Vogel[56]. This stereoselective cycloaddition established the second oxygen functionality and could be scaled up to 100 g without erosion of yield or stereoselectivity. Quinine-mediated regioselective methanolysis[58] of anhydride 11 resulted in the methyl ester and acid 12. Subsequently, a stereospecific Upjohn *exo*-dihydroxylation[59] of the olefin established the third and the fourth oxygen functionalities (with simultaneous 1,2-diol protection) and produced the mono-acid 13, whose structure was confirmed by X-ray crystallographic analysis of the single crystal (CCDC#: 2184304).

Decarboxylative hydroxylation was carried out to introduce the fifth oxygen functionality at the C5 position. Initially, high-valent metal reagents were examined as oxidants but were inadequate owing to substrate decomposition. Mild radical conditions, including Barton or organophotoredox-promoted decarboxylation in the presence of a radical initiator and oxygen under UV irradiation[60–62], were also unsuccessful (Fig. 3b). After considerable experimentation, a Ru-catalyzed photoredox decarboxylative hydroxylation[63] of the N-hydroxyphthalimide (NHPI) ester of 13 produced 14 as a single detectable diastereomer in 66% yield, albeit with an inverted configuration at C5 relative to TTX. Previous syntheses[8,20] revealed that steric

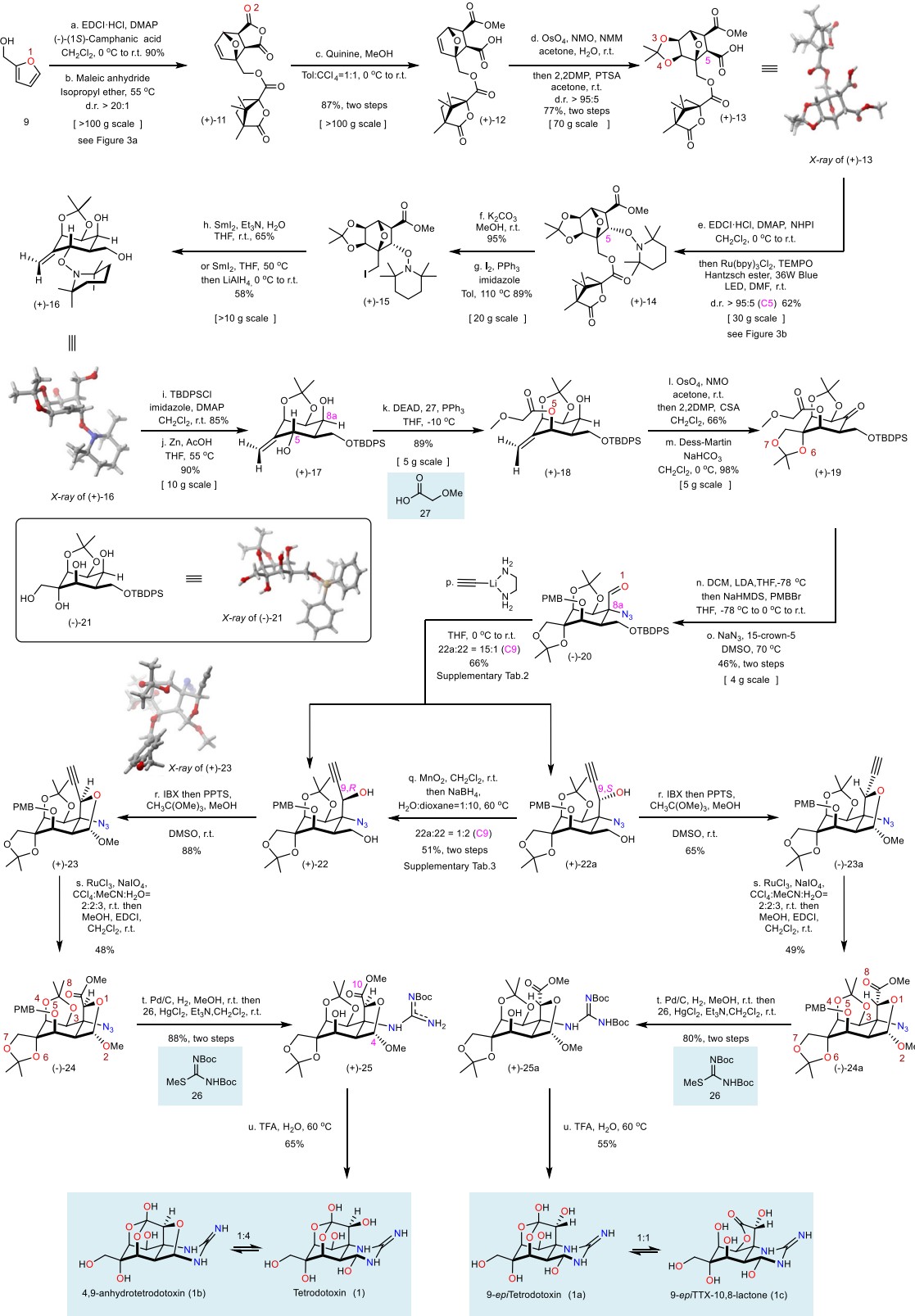

**Fig. 2 | Completion of total syntheses of TTX and 9-*epi*TTX.** The oxygen atoms are numbered and highlighted in red when they are constructed, and the nitrogen atoms are highlighted in blue; carbon atoms numbered 4, 5, 8a, 9, and 10 are indicated in pink. EDCI·HCl 1-ethyl-3-(3-dimethylaminopropyl) carbodiimide hydrochloride, DMAP 4-dimethylaminopyridine, Tol toluene, NMO

N-methylmorpholine N-oxide, NMM N-methylmorpholine, 2,2DMP 2,2-dimethoxypropane, NHPI N-Hydroxyphthalimide, DEAD diethyl azodicarboxylate, CSA camphorsulfonic acid, DCM dichloromethane, PPTS pyridinium *p*-toluenesulfonate, TFA trifluoroacetic acid.

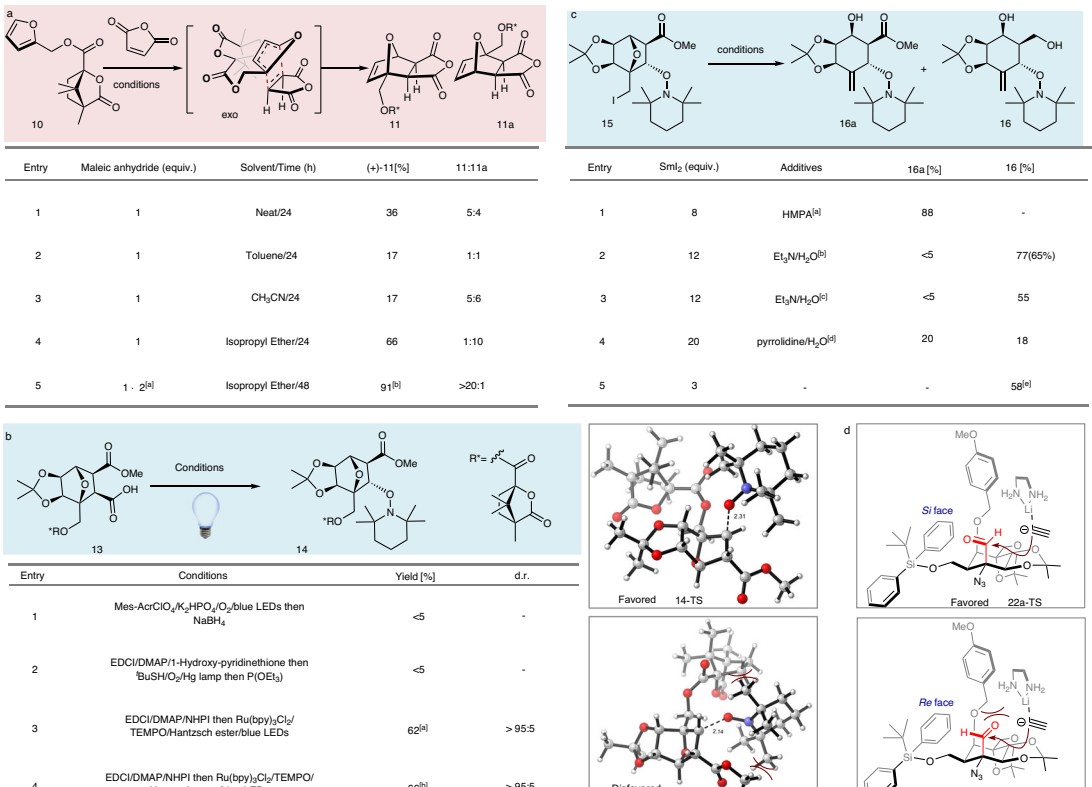

**Fig. 3 | Optimization of reaction conditions.** Unless otherwise stated, the yields were determined by ¹H NMR with 1,3,5-trimethoxybenzene as the internal standard. **a** Optimization of the Diels−Alder reaction. [a] After 12 h, the same equiv. maleic anhydride was added. [b] Scale up to 100 g, isolated yield with 6% (molar ratio) maleic anhydride. **b** Optimization of decarboxylative hydroxylation. [a] d.r. > 95:5, 30 g scale. [b] Isolated yield on 1 g scale using a circulating flow system. **c** Optimization of the SmI₂/H₂O/amine-mediated fragmentation. [a] HMPA (10 eq).

[b] Et₃N (24 eq)/H₂O (24 eq). The yield in the bracket is an isolated yield. [c] Et₃N (36 eq)/H₂O (36 eq). [d] pyrrolidine (60 eq)/H₂O (60 eq). [e] SmI₂ (3 eq), 55 °C, without purification followed by reduction using LiAlH₄. The yield is isolated yields for the two steps on decagram scale. **d** Selectivity of nucleophilic addition. The aldehyde functional group is highlighted in red. N.D. not determined, MTBE methyl *tert*-butyl ether.

hindrance at the C5 position is troublesome for the following functional group manipulations. Therefore, we utilized a late-stage con-figurational inversion strategy to simplify the stereoselective oxygen functionality installation sequence. Notably, this photoredox dec-arboxylative hydroxylation could also be scaled up by employing cir-culating flow photochemistry without compromising the yields or diastereoselectivity (entry 4, Fig. 3b). To interpret the diastereoselec-tivity and analyze the steric effect of this radical addition; we per-formed the density functional theory (DFT) calculations. The DFT calculations supported a clear radical addition preference for the experimentally observed stereoisomer at C5 that stemmed from the radical addition from the convex face of the oxo-bridge ring. (ΔΔ$G$ = 3.4 kcal/mol and predicted dr > 99:1; details of computation results are shown in the Supplementary Information.)

With compound 14 in hand, we investigated the functional group interconversions of this oxo-bridge ring system. The auxiliary (−)-camphanic acid was first removed by transesterification with methanol, providing the primary alcohol, which was then subjected to an Appel reaction, giving the alkyl iodide 15. The chiral auxiliary could be recycled as methyl camphanate. A variety of reductive conditions applied to the alkyl iodide 15 failed to produce the desired oxo-bridge ring-opening product. After intensive exploration of the reductive conditions, we developed a successful reaction sequence (Fig. 3c): the initial SmI₂ mediated single electron transfer homolytically cleaved the carbon-iodide bond and generated a primary carbon radical, which could be further reduced by Sm(II) to a carbanion[64,65] to drive the bridged C−O bond cleavage. The primary alcohol was generated from concurrent methyl ester reduction by SmI₂, while the N-O bond of

TEMPO remained unaffected due to steric hindrance. In the presence of hexamethylphosphoramide (HMPA), only intermediate 16a was obtained without reduction of the methyl ester to diol 16 (entry 1, Fig. 3c). Activation of SmI₂ with H₂O and Et₃N in a 1:2:2 ratio created a stronger reductant[66], which allowed for the reduction of the methyl ester (entry 2) in a 77% yield as determined by ¹H NMR. Increasing amounts of H₂O and Et₃N or replacing Et₃N with pyrrolidine resulted in complex product mixtures (entries 3 and 4). The procedure could also be modified to a two-step protocol involving fewer equivalents of SmI₂ to afford 16a, followed by a LiAlH₄ reduction to give 16 in 58% yield on a decagram scale (entry 5). The relative configuration of 16 was verified by X-ray crystallography of the single crystal (CCDC#: 2182018).

The construction of azidoaldehyde 20 started with selective protection of the primary alcohol in 16 using the sterically hindered TBDPSCl. The N-O bond of TEMPO in the resulting alkene was reduc-tively cleaved with Zn powder, giving the allylic alcohol 17. The incorrect configuration of C5-OH was then inverted by a chemoselec-tive Mitsunobu reaction of C5 allylic alcohol in the presence of free C8a secondary alcohol with 2-methoxyacetic acid 27, delivering the fifth oxygen functionality in 18 in excellent yield. Other acids, such as acetic acid or benzyloxyacetic acid, afforded products with low yields. The sixth and seventh oxygen functionalities were established via a dia-stereoselective Upjohn dihydroxylation followed by protection as the acetonide, whose relative configuration was confirmed by X-ray crys-tallography of the derivative 21 (CCDC#: 2184298) (See the supple-mentary information). The secondary alcohol underwent Dess-Martin oxidation to afford the ketone 19 in excellent yield. An intramolecular Mannich reaction between the α position of methoxyacetic acid and

the ketone 19-derived imine was unfeasible. The intermolecular nucleophilic addition of a variety of nucleophiles also exclusively produced a diastereomer with the wrong configuration at C8a (Supplementary Fig. 2a). Although Darzens reaction of 19 with $\alpha$-haloester smoothly generated an $\alpha$, $\beta$-epoxy ester (glycidic ester), the stereoselective and regioselective epoxide opening strategy proved unfruitful in the presence of different types of nitrogen-based nucleophiles (Supplementary Fig. 2b). The nucleophilic addition of Sato's dichloromethyllithium (LiCHCl$_2$) to the ketone 19 was successfully afforded the spiro $\alpha$-chloroepoxide as a single diastereomer[22,54] and concurrently removed the ester group at C5−OH, which was protected with a p-methoxybenzyl group (PMB) in one pot. Regioselective epoxide opening of the resulting chloroepoxide with NaN$_3$ proceeded smoothly to afford the $\alpha$-azido aldehyde 20 on a gram-scale, with the correct configuration of the C8a quaternary stereogenic center.

With the construction of the highly oxygen-substituted carbocyclic core 20 accomplished, we began to address the synthetic challenge of constructing the complex dioxa-adamantane core and the guanidinium hemiaminal moieties.

The $\alpha$-azido aldehyde 20 was subjected to a 1,2-addition with lithium acetylide (Supplementary Table 2), followed by the removal of the TBDPS group to produce two diastereomers (22/22a = 1:15) that could undergo divergent synthesis to both TTX and 9-epiTTX. Presumably owing to the steric hindrance introduced by the bulky TBDPS and PMB groups, the lithium acetylide preferentially attacked from the less sterically hindered si face and generated the undesired diastereomer 22a (Fig. 3d). Extensive exploration of the reaction conditions revealed that the stereochemistry of C9 of 22a could be inverted in a 2:1 ratio (22/22a = 2:1) via sequential MnO$_2$-mediated chemoselective oxidation followed by NaBH$_4$ reduction (Supplementary Table 3). IBX oxidation of the primary alcohol 22 provided the corresponding bridged hemiacetal, which was converted to the acetal 23 with trimethylorthoacetate. The structure and the stereochemistry of 23 were confirmed by single-crystal X-ray crystallography (CCDC#: 2184305). Distinct from previous syntheses that heavily focused on the lactone formation between C5−OH and the C10−COOH as the advanced intermediate, our strategy pinpointed the issue of conformational control for precise functional group manipulations on the stereochemically complex framework. Decreasing the conformational flexibility by the bridged tetrahydrofuran acetal ring formed between C9 and C4 is critical to the efficiency of the following transformations, including alkyne oxidative cleavage, guanidine installation, and one-step cyclic guanidinium hemiaminal and orthoester formation, thus demonstrating a unique and concise strategy for the final stage of TTX synthesis.

Oxidative cleavage of alkyne 23 with RuCl$_3$/NaIO$_4$ followed by esterification afforded methyl carboxylate 24. Simultaneous PMB deprotection and azido reduction by hydrogenation efficiently delivered the tertiary amine, which was guanidinylated[67] in situ with bis-Boc protected isothiourea 26, leading to the penultimate intermediate 25. To our delight, treatment of this unprecedented compound 25 with trifluoroacetic acid at 60 °C afforded a global deprotection and successfully installed both the hemiaminal and the orthoester of TTX, leading to the final product TTX (1) and 4,9-anhydroTTX (1b) in a 1:1 mixture. The use of 2% TFA-d in deuterium oxide further converted this mixture to a 4:1 ratio favoring TTX (see supplementary information)[8]. A similar synthetic process was used to convert the diastereomer 22a to the final 9-epiTTX (1a) and its 10,8-lactone form (1c) in 5 steps (14% overall yields). The spectroscopic data ($^1$H NMR, $^{13}$C NMR, HRMS) of synthetic TTX and 9-epiTTX were identical to those of the authentic reference samples[7–9].

## Voltage-gated sodium channels blocking experiment

TTX in most biomedical studies is a mixture of the ortho ester, the lactone form, and 4,9-anhydroTTX[8,11]. To investigate the biological activities of a pure TTX, we synthesized and purified a single form of TTX (S) from the methyl carboxylate 24 according to Fukuyama's strategy[24]. Another sample named TTX (C) (purchased from Tocris Bioscience, the ratio of TTX to 4,9-anhydroTTX is 10:3 as analyzed by $^1$H NMR, purity > 99%, Supplementary Fig. 3) was utilized for comparison.

To evaluate the potency of TTX (S) to block sodium channels, we performed a series of electrophysiological experiments. First, we compared the blocking capability of two different sources of TTX on a single subtype of sodium channel on human HEK-Na$_v$1.7 cells. We measured the voltage-dependent Na$_v$1.7 currents in the absence or presence of TTX. We found that TTX (1 µM) from different sources almost completely blocked Na$_v$1.7 currents (Fig. 4a−f). To compare the potency of TTX(S) and TTX (C), we systematically measured voltage-dependent Na$_v$1.7 currents at different concentrations of TTX (Fig. 4g−i). For the Na$_v$1.7 currents evoked by membrane depolarization from −80 mV to +10 mV, the IC50s of TTX (S) and TTX (C) were approximately 2.69 nM and 3.78 nM, respectively. These data suggested that the potency of TTX (S) to block sodium currents may be higher than that of TTX (C). Second, we measured the voltage-dependent Na$_v$1.5 currents in the absence or presence of TTX (1 µM). We found that both TTX (C) and TTX (S) blocked voltage-dependent Na$_v$1.5 currents (Supplementary Fig. 4). Finally, we examined whether TTX (S) and TTX (C) block sodium currents evoked by voltage ramps (−70 mV to + 10 mV) in primary hippocampal neuronal cultures. We found that our synthetic pure TTX (S) also exhibited a stronger effect in blocking the sodium currents amplitude than TTX(C) (Supplementary Fig. 5).

In summary, we have achieved the asymmetric synthesis of 9-epiTTX (1a) (22 steps) and one of the shortest syntheses of TTX (1) (24 steps, following the Rules for Calculating Step Counts[68,69].) from the easily accessible furfuryl alcohol. The hundred-gram-scale asymmetric preparation of cyclohexane (+)-12 showcases the power of the stereoselective Diels-Alder reaction in the scale-up synthesis of a carbocyclic ring with a dense array of functionalities[70]. The precise introduction of the oxygen functionality at the C-5 position via photochemical decarboxylative hydroxylation highlights the advance of free radical transformation performed on a sterically demanding carbocyclic skeleton. The SmI$_2$-mediated sequential reactions of reductive fragmentation, oxo-bridge ring opening, and ester reduction, followed by diastereoselective Upjohn dihydroxylation enable a gram-scale synthesis of highly oxidized intermediate (+)-19. The bridged tetrahydrofuran acetal setting simplifies the endgame and facilitates the rapid formation of the cyclic guanidinium hemiaminal and orthoester in one pot. Notably, the present synthesis served as a testbed for precise functional group manipulations on the densely functionalized and stereochemically complex frameworks and should be readily applicable to the synthesis of other heavily oxygenated polycyclic natural products. The concise synthetic strategy is suitable for the production of TTX congeners or derivatives that support further pharmacology investigations and should be amenable to large-scale synthesis of TTX for analgesic drug development, particularly for non-opioid cancer pain treatment.

## Methods
### Culture of primary hippocampal neurons
C57BL/6 J mice strain was maintained in an animal facility with 12 h light/12 h dark cycles, temperature (22–24 °C), and humidity (40–60%) at the National Institute of Biological Sciences, Beijing. PO pups were used for dissociating primary hippocampal neurons. Sex was not considered in the study design because there was no sex correlation involved in this experiment.

Hippocampi were isolated from P0 wildtype mice of either sex (C57BL/6 J) and kept in ice-cold Hank's balanced salt solution (Sigma) and incubated with papain (10 U/ml in 1 mM CaCl$_2$ and 0.5 mM EGTA)

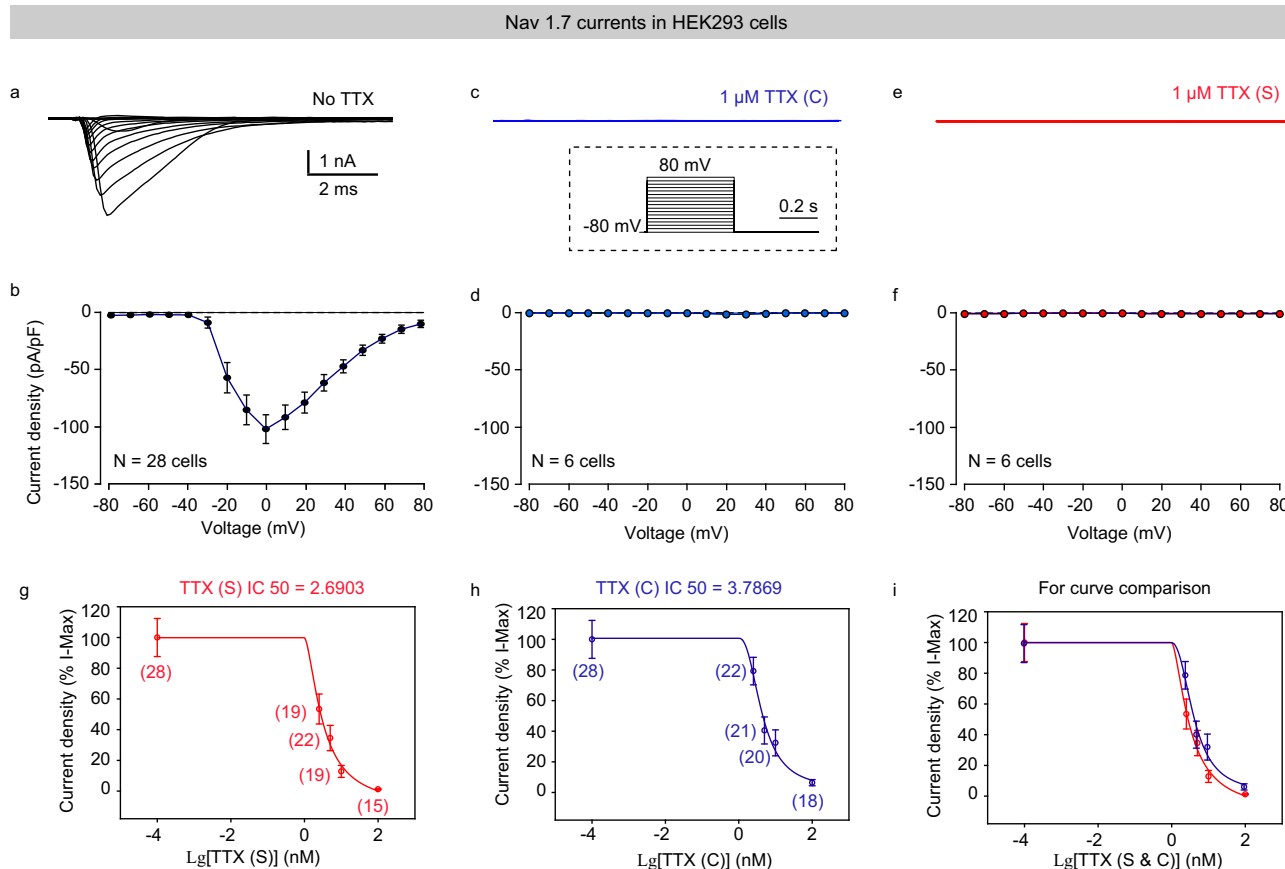

**Fig. 4 | The blocking potency between TTX(S) and TTX(C) on HEK-Na$_v$1.7 cells.**
**a–f** Representative traces (top) and quantitative analyses (bottom) of HEK-Na$_v$1.7 currents under the conditions of No-TTX (**a, b**), 1 μM TTX(C) (**c, d**), and 1 μM TTX(S) (**e, f**). Inset in (**c**), a schematic image showing the protocol of step-wise voltage-clamp from −80 mV to +80 mV in 10 mV increments. **g–i** Quantitative analyses of concentration-dependent Na$_v$1.7 currents evoked by membrane depolarization from −80 mV to 0 mV in the presence of TTX (S) (**g**) and TTX (C) (**h**). For a comparison of these two curves, see (**i**). Note: The IC50s of TTX (S) and TTX (C) were approximately 2.6903 nM and 3.7869 nM, respectively. The number of cells in each group is shown in brackets in (**g–i**). Error bars in **b, g–i** indicate mean ± SEM. TTX(S) indicates synthetic TTX; TTX(C) indicates commercialized TTX from Tocris Bioscience Inc.

at 37 °C for 20 min. Cells were dissociated by 1 mL pipette. Dissociated cells were plated on Matrigel-coated circular glass coverslips (12 mm diameter) in 24-well dishes and cultured for 14–16 days in vitro (DIV) in 1 ml of MEM (Invitrogen) supplemented with B27 (Invitrogen), glucose, transferrin, fetal bovine serum, and AraC (Sigma).

Animal experimentation: Animal care and use followed the institutional guidelines of the National Institute of Biological Sciences (NIBS), Beijing (Approval ID: NIBSLuoM15C), and the Regulations for the Administration of Affairs Concerning Experimental Animals of China.

### Electrophysiological recording of cultured hippocampal neurons

The bath solution contained the following (in mM): 140 NaCl, 5 KCl, 2 MgCl$_2$, 2 CaCl$_2$, 10 HEPES, and 10 glucose, adjusted to pH 7.4 with NaOH. The whole-cell intracellular pipette solution contained the following (in mM): 126 CsMeSO$_3$, 10 HEPES, 1 EGTA, 0.1 CaCl$_2$, 3 GTP-Na3, 4 ATP-Mg, 8 Na2-phosphocreatine, adjusted to pH 7.4 with CsOH. The current signals were recorded with MultiClamp 700B and Clampex 10 data acquisition software (Molecular Devices). The resistance of pipettes varied between 4.0–7.0 MΩ. Recordings with series resistances of >15 MΩ were rejected. Neurons were clamped at −70 mV in the presence of TTX (S) or TTX (C). Sodium currents are evoked by a 50-ms width ramp voltage from −70 mV to 10 mV. Leakage currents of >400 pA were rejected. Data were sampled at 10 kHz. The data were analyzed using Clampfit 10.5 (software) and Prism 6.02 (GraphPad

Software). Sodium current amplitudes were analyzed after being normalized to the sodium current amplitude of the 1 μM TTX Compounds group.

### Voltage-gated sodium channel assay

HEK-293 cells stably expressing Na$_v$ 1.5 (human) or Na$_v$ 1.7 (human) were donated by Fan Zhang lab in Hebei Medical University and cultured in Dulbecco's Modified Eagle Medium (DMEM, Gibco) containing 4.5 mg/ml glucose, 10% fetal bovine serum (FBS, Gibco), 100 U/ml penicillin, 100 μg/mL streptomycin and incubated at 37 °C with 5% CO$_2$. When cell confluency reached 70%, the cells were treated with 0.05% trypsin (Gibco) and put on poly-D-lysine (Sigma)-coated 12 mm coverslips for whole-cell electrophysiological characterization.

Sodium current signals were recorded using MultiClamp 700B, Clampex 10 data acquisition software (Molecular Devices), and glass micropipettes (4–7 MΩ) in stably expressing Na$_v$ 1.5 (human) or Na$_v$ 1.7 (human) HEK-293 cells. The data were analyzed using Clampfit 10.5 (software) and Prism 6.02 (GraphPad Software). For recording the voltage-dependent currents, the electrodes were filled with the internal solution composed of (in mM) 40 CsCl, 10 NaCl, 10 EGTA, 105 CsF, 10 HEPES, pH 7.3 with CsOH. The extracellular solution was composed of (in mM) 130 NaCl, 4 KCl, 1 MgCl$_2$, 1.5 CaCl$_2$, 5 D-Glucose monohydrate, 5 HEPES, pH = 7.4 with NaOH. The voltage dependence of ion current was elicited using a protocol consisting of steps from a holding potential of −80 mV to voltages ranging from −80 to

80 mV for 500 ms in 10 mV increments. The current density is obtained by dividing the current amplitude by the membrane capacitance and plotted against the voltage. Data were sampled at 10 kHz. Recordings with series resistances of >15 MΩ or leakage currents of >400 pA were rejected. The function (The formula: $xb$ = IC50 * $10^{((1/Hillslope)\, *\, \log(2^{(1/s)} - 1))}$; $f1$ = min + (max − min)/ $(1 + (xb/x)^{Hillslope})^s$; $f$ = if $(x < =0$, if (Hillslope > 0, min, max), $f1$)) in SigmaPlot 12.0 software were used for curve fitting and calculation of IC50.

### Reporting summary
Further information on research design is available in the Nature Portfolio Reporting Summary linked to this article.

### Data availability
The X-ray crystallographic coordinates for structures reported in this study have been deposited at the Cambridge Crystallographic Data Centre (CCDC), under deposition numbers 2184304 (13) (https://doi. org/10.5517/ccdc.csd.cc2c9yf2), 2182018 (16) (https://doi.org/10.5517/ ccdc.csd.cc2c7kpw), 2184298 (21) (https://doi.org/10.5517/ccdc.csd. cc2c9y7w), 2184305 (23) (https://doi.org/10.5517/ccdc.csd.cc2c9yg3). Copies of the data can be obtained free of charge via https://www.ccdc. cam.ac.uk/structures/. All other data supporting the findings of this study, including experimental procedures and compound characterization, NMR, and HPLC, are available within the Article and its Supplementary Information or from the corresponding author upon request. The raw NMR data and HPLC traces, optimized coordination data for the calculated structures, and data of electrophysiological experiments are available at figshare under accession code https://doi. org/10.6084/m9.figshare.23291687 under the Creative Commons Attribution 4.0 International license. Source data are provided in this paper.

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

## Acknowledgements

This work was supported by the National Natural Science Foundation of China (grant nos. 21971018 and 82225041, X.Q.). The authors gratefully acknowledge the Beijing Municipal Government and Tsinghua University for their financial support. We thank Prof. Joseph Ready and Prof. Uttam Tambar for their scientific comments and thank Drs. Jianwei Bian, Bo Liu, Shuanhu Gao, and Weiqing Xie for crucial suggestions.

## Author contributions

X.Q. conceived the study; P. Chen, J.W., and X.Q. designed the synthetic route and prepared the manuscript; P. Chen and J.W. carried out most of the chemical synthesis and prepared the supplemental information; P.

Chen, J.W., Y.W., Y.S., and Q.W. analyzed the data; S.Z., X.C., and P. Cao performed the biological study, S.B. performed the DFT computation. All authors discussed the results and commented on the paper.

## Competing interests

The authors (P. Chen, J. W., Y. W., Y. S., Q. W., X. Q.) declare the following competing interests: a patent application based on this study (WIPO Application No. PCT/CN2022/111861). The remaining authors (S. Z., S.B. X.C., P. Cao) declare no competing interests.
