## [Peer Review File · Nature Communications]

REVIEWER COMMENTS

Reviewer #1 (Remarks to the Author):

Qi and colleagues describe a short synthesis of TTX, a legendary synthetic target, and, for the first time, of 9-epi TTX. There is indeed a need to procure TTX through scalable and efficient chemical syntheses once the substance gets approved as a drug (clinical trials are ongoing). The value of the paper lies in the efficient synthetic strategy and the potential scalability. It is very nice work, suitable for Nature Communications, given the complexity of the highly polar and densely functionalized target and the novelty and stereoselectivity of the 9-epi TTX synthesis. The work of others in the field, such as Sato's, has been properly acknowledged and the endgame is very different.

The diastereoselectivity of the opening Diels-Alder reaction with such a remote auxiliary is indeed remarkable. The synthesis has two undesired stereoselectivities with regards to TTX: 1) in the oxidative decarboxylation, which could be nicely corrected by Mitsunobu reaction, and 2) in the acetylide addition installing the C9 stereocenter. The solution for the latter is less satisfactory but, since 9-epi TTX is now a natural product, there is a silver lining to this.

Overall, this a very good synthesis of a highly important molecule that continues to fascinate and challenge synthetic chemists, is an indispensable tool in neuroscience, and is at the verge of clinical approval. I am supportive of publishing it in Nature Communications.

Reviewer #2 (Remarks to the Author):

This is an interesting paper that describes a novel process for synthesizing TTX and possibly other heavily oxygenated polycyclic natural products. TTX is difficult to synthesize, so this process is an important technical advance. However, it is not clear how easily this procedure can be modified for other difficult compounds. The functional analysis is also concerning. The sodium currents in primary neuronal cultures are difficult to voltage-clamp due to space-clamp limitations. The data shown in figure 4 suggests that the synthetic TTX has an IC₅₀ around 22 nM, which is consistent with other various estimates for the IC₅₀ of TTX on a multitude of TTX-sensitive voltage-gated sodium channels. Figure 4 compares the newly synthesized TTX to a "commercial" TTX. The commercial TTX seems to have reduced potency (IC₅₀ ~45 nM). However, it is unclear what the source of this is and how representative it is of commercial TTX. It is important to test the synthetic TTX on cells that can be adequately voltage- and space clamped and to generate more complete characterization of the

synthetic TTX on several different sodium channel isoforms. As presented, the study represents a technical advance but not a significant advance that would likely appeal to a broad readership.

Minor comments

The following statement "Extensive pharmacological investigations, including clinical trials, have demonstrated the immense promise of TTX in pain treatment and detoxification from heroin addiction" is not fully supported by the single reference cited. Many would argue that while TTX has some promise as a therapeutic, describing it as "immense promise" is not justified.

Reviewer #3 (Remarks to the Author):

The manuscript reports a very elegant total synthesis of tetrodotoxin and 9-epi-tetrodotoxin. Given that the great synthetic challenges of the target molecules and their significant bio-functions, the synthesis reported will certainly be great news to natural product chemists, synthetic chemists, medicinal chemists as well as chemical biologist. Recently, Trauner has reported a concise total synthesis of tetrodotoxin (*Science* 377, 411-415 (2022)). From synthetic chemistry point of view, the current synthesis is, if not better, at least at the same quality as Trauner's synthesis. I am thrilled to see the flourish in the field of TTX by the synthetic progress. I enthusiastically recommend the publish of the work in *Nature Communication*. It is the type of the manuscript your journal will not want to miss. Chemistry wise, the authors have done a great job. I do not have suggestion or questions. I only have some suggests regarding the presentation of the manuscript.

1) Figure 1 summarized previous total synthesis of TTX. But I do not see Kishi's pioneer work. This is somehow regretful considering the leading role of Kishi in the field. I would like suggest authors to solute to Professor Kishi by adding some space to show the respect.

2) As another synthetic target, structure of 1a in figure 1 should also be highlighted with a light blue background.

3) Regarding the formation of 1b in figure 2, the arrows authors used implied a SN2 pathway, which is negotiable. In addition, the arrows depicting the formation of ortho ester is not ideal either. A simple solution is that authors just delete this structure in their final draft of the manuscript.

Response letter

REVIEWER COMMENTS

Reviewer #1 (Remarks to the Author):

Qi and colleagues describe a short synthesis of TTX, a legendary synthetic target, and, for the first time, of 9-*epi*TTX. There is indeed a need to procure TTX through scalable and efficient chemical syntheses once the substance gets approved as a drug (clinical trials are ongoing). The value of the paper lies in the efficient synthetic strategy and the potential scalability. It is very nice work, suitable for Nature Communications, given the complexity of the highly polar and densely functionalized target and the novelty and stereoselectivity of the 9-*epi* TTX synthesis. The work of others in the field, such as Sato's, has been properly acknowledged and the endgame is very different.

The diastereoselectivity of the opening Diels-Alder reaction with such a remote auxiliary is indeed remarkable. The synthesis has two undesired stereoselectivities with regards to TTX: 1) in the oxidative decarboxylation, which could be nicely corrected by Mitsunobu reaction, and 2) in the acetylide addition installing the C9 stereocenter. The solution for the latter is less satisfactory but, since 9-*epi* TTX is now a natural product, there is a silver lining to this.

Overall, this a very good synthesis of a highly important molecule that continues to fascinate and challenge synthetic chemists, is an indispensable tool in neuroscience, and is at the verge of clinical approval. I am supportive of publishing it in Nature Communications.

Our Response:

We appreciate the thorough summary and professional viewpoints from this reviewer, and we are especially thankful for his/her strong supports!

Reviewer #2 (Remarks to the Author):

This is an interesting paper that describes a novel process for synthesizing TTX and possibly other heavily oxygenated polycyclic natural products. TTX is difficult to synthesize, so this process is an important technical advance.

Our Response:

We appreciate the overall positive judgments from this reviewer!

Reviewer #2: However, it is not clear how easily this procedure can be modified for other difficult compounds.

Our Response:

Thanks for pointing out this great question. We totally understand this reviewer's concern, especially for scalable synthesis, which is the advance of our synthesis and our final goal. For the synthetic generality, there are several notable features of our synthetic route that we believe could be modifiable for the synthesis of TTX analog and other heavily oxygenated polycyclic natural products:

- I. The highly heteroatoms-substituted and pseudo-symmetric cyclohexane skeleton was assembled via a stereoselective Diels-Alder reaction and a chemoselective cyclic anhydride opening strategy. **(This process is suitable for the enantioselective synthesis of highly-substituted carbocyclic ring system)**
- II. A free radical decarboxylative hydroxylation was applied to install the oxygen at C-5, and circulating flow photochemistry was developed for gram-scale synthesis, **demonstrating radical chemistry's strategic value in functional group interconversions on highly steric hindered and complex settings.**
- III. An innovative SmI_2 -mediated cascade reaction of radical elimination, oxo-bridge ring opening, and ester reduction, followed by diastereoselective Upjohn dihydroxylation and α -chloroepoxidation enabled a gram-scale synthesis of the densely oxidized cyclohexane core and concurrently constructed the continuous chiral centers on this highly sterically hindered skeleton, which represents a **practical strategy for the assembly of saturated and heteroatom-enriched heterocyclic molecules.**
- IV. The final TTX and *epi*-TTX were generated from the mutual intermediate 22 that can be selectively modified at several positions based on the different functionalities, such as alkyne, **free primary alcohol**, **free secondary alcohol**, **azide**, **PMB-protected-alcohol** and **dimethylacetyl protected-diol**. All these functional groups could be modified selectively and derived to TTX analogs (see Scheme below. The same color of the functional groups on starting material and product represents the "functional group interconversion" between these FGs).

Reviewer #2: The functional analysis is also concerning. The sodium currents in primary neuronal cultures are difficult to voltage-clamp due to space-clamp

limitations. The data shown in figure 4 suggests that the synthetic TTX has an IC50 around 22 nM, which is consistent with other various estimates for the IC50 of TTX on a multitude of TTX-sensitive voltage-gated sodium channels. Figure 4 compares the newly synthesized TTX to a “commercial” TTX. The commercial TTX seems to have reduced potency (IC50 ~45 nM). However, it is unclear what the source of this is and how representative it is of commercial TTX. It is important to test the synthetic TTX on cells that can be adequately voltage- and space clamped and to generate more complete characterization of the synthetic TTX on several different sodium channel isoforms. As presented, the study represents a technical advance but not a significant advance that would likely appeal to a broad readership.

Our Response:

We really appreciate the constructive suggestions from this reviewer! We agree with the reviewer and conducted new functional studies accordingly using human HEK-293T cells, which stably express either Na_v1.5 or TTX-sensitive Na_v1.7.

Firstly, we want to clarify the issue of the commercial TTX, which was purchased from Tocris bioscience (Batch NO: 51A, Purity > 99%). We agree with this reviewer that the commercial-TTX we tested is not a standard or representative sample for commercial samples. So, we added the criteria of quantitative ratio of TTX to 4,9-anhydro TTX and purity to represent the samples from commercial sources.

Secondly, to solve the important problem of “*The sodium currents in primary neuronal cultures are difficult to voltage-clamp due to space-clamp limitations*”, we generated human HEK-293T cells, which stably express either TTX- Na_v1.5 or TTX-sensitive Na_v1.7. HEK-293T cells are widely used for the study of ion channel currents by transfection with plasmid vectors, which express the corresponding ion channels.

(For the references, please see: *British Journal of Pharmacology*, 2014, 171(4), 1054-1067.; For the study of Na_v1.5, please see: *Circulation research*. Hagerstown, MD: Lippincott Williams & Wilkins, 2020, 127(Suppl_1), AMP113-AMP113.) For the study of Na_v1.7, please see: *Science*, 2019, 363(6433), 1303-1308. 10.1126/science.aaw2493.)

Next, we compared the blocking capability of two different sources of TTX on a single subtype of sodium channel on human HEK-Na_v1.5 cells or HEK-Na_v1.7 cells (Fig. R1 and Fig. R2). The Na_v1.7 channel is a TTX-sensitive voltage-gated sodium channel. We detected the normal HEK-Na_v1.7 currents under no-TTX condition by voltage clamp from -80 mV to 80 mV (Fig. R1 A). The HEK-Na_v1.7 currents could be successfully induced and was completely blocked by 1 μM TTX(C) (Fig. R1 B&C). The results suggest that the HEK-Na_v1.7 cells can be used to measure the blocking ability of different sources of TTX and TTX

analog.

We then measured the HEK-Nav_v1.7 currents of HEK-Nav_v1.7 cells that were treated with TTX(S) and TTX(C) under the same conditions. Our results indicated that TTX(S) has better blocking efficiency than TTX(C) (Fig. R1 D). Further, we determined the blocking efficiency between TTX(S) and TTX(C) on HEK-Nav_v1.5 cells. The Nav_v1.5 channel was previously identified as TTX-resistant voltage-gated sodium channel (*Trends in pharmacological sciences*, 2005, 26(10), 496-502.) (*Journal of Neuroscience*, 2017, 37(20), 5204-5214.). We also detected HEK-Nav_v 1.5 currents under no-TTX condition or 1 μM TTX(S) condition or 1 μM TTX(C) condition by voltage clamp from -80 mV to 80 mV (Fig. R2 A, B, C). The evoked HEK-Nav_v 1.5 currents could be partially blocked by both 1 μM TTX(S) and 1 μM TTX(C) and TTX(S) has slightly better blocking efficiency than TTX(C).

Furthermore, we compared the quantitative currents densities. The bar graph shows that both TTX(S) and TTX(C) display a significant decrease in HEK-Nav_v1.5 currents density and slightly lower currents density was observed for 1 μM TTX(S) than 1 μM TTX(C) (Fig. R2 D).

Figure R1 The TTX(S) has a stronger blocking effect on HEK-Nav_v1.7 currents than TTX(C). (A) A schematic image represents an evoked step-wise protocol by voltage clamp on HEK-

Na_v1.7 cells from -80 mV to 80 mV in 10-mV increments. (B) Representative traces of HEK-Na_v1.7 currents under no-TTX condition or 1 μM TTX(C) condition. (C) The plot graph exhibits the HEK-Na_v1.7 currents densities evoked by series of clamped voltage from -80 mV to 80 mV under no-TTX condition. N= 8 cells. Error bar, Mean ± SEM. (D) Pie charts display the number of cells successfully induced to emit currents under different conditions with different concentrations of TTX treatment. The left of pie charts shows the treatment conditions, and the bottom of pie chart shows the total cell number.

Figure R2 (A) A schematic image represents an evoked step-wise protocol by voltage clamp on HEK-Na_v1.5 cells from -80 mV to 80 mV in 10-mV increments. (B) Representative traces of HEK-Na_v1.5 currents under no-TTX condition or 1 μM TTX(S) condition or 1 μM TTX(C) condition. (C) The plot graph displays the HEK-Na_v1.5 currents densities evoked by series of clamped voltage from -80 mV to 80 mV under no-TTX condition or 1 μM TTX(S) condition or 1 μM TTX(C) condition. Control group, N = 10 cells; 1 μM TTX(S) group, N = 11 cells; 1 μM TTX(C) group, N = 10 cells. (D) The bar graph displays quantitative comparison of HEK-Na_v1.5 currents densities at several clamped voltages under different TTX treatment. Control group, N = 10 cells; 1 μM TTX(S) group, N = 11 cells; 1 μM TTX(C) group, N = 10 cells. Error bar, Mean ± SEM. T.Test, ** *p* value < 0.01 .

Besides the above cell assay, we also conducted additional structural analysis:

One notable distinction between "synthetic" TTX and "commercial" TTX is the relative concentration of 4,9-anhydro TTX. Previous studies (*Am J Physiol Cell Physiol* 293, C783–C789, 2007) have examined the activity of 4,9-anhydro TTX, which has exhibited distinct selectivity in inhibiting voltage-gated sodium

channels compared to TTX. In our investigation, we evaluated the effects of our "synthetic" and "commercial" TTX on Na_v1.5, and Na_v1.7 channels to confirm the lower blocking ability of "commercial" TTX is attributed to the negative effect from 4,9-anhydro TTX. All these results suggest that modifications at the C9 position might play a significant role in identifying various voltage-gated sodium channel isoforms. To further investigate the mechanism of action of TTX, 9-*ep*/TTX, and 4,9-anhydro TTX, we performed structure-based in-silico binding studies of 4,9-anhydro TTX and 9-*ep*/TTX.

Molecular dynamics simulations in combination with MM/GBSA binding energy calculations for TTX, 4,9-anhydro TTX, and 9-*ep*/TTX binding the voltage-gated sodium channel 1.7 isoform were employed. Our initial structure was derived from the Cryo-EM structure of the Na_v1.7-TTX complex (PDB code: 7w9m). We pruned the original structure and restrained the alpha-carbons of the protein backbone beyond a distance of 13 Å from the TTX binding site during the simulation (Fig. R3) to focus specifically on the pore moiety(blue).

Figure R3: Blue: Pruned Na_v1.7 protein after 300ns restrained simulation. Cyan: Cryo-EM structure of Na_v1.7-TTX complex (PDB code: 7w9m).

The results are presented in Fig. R4. Throughout the 300 ns simulations, the protein backbones reached equilibrium and maintained their stability, with a Root Mean Square Deviation (RMSD) of less than 2 Å (Fig. R4 A, C, E). During the simulation, TTX exhibited two distinct poses. These poses were relatively similar, with binding enthalpies of -67.87 ± 6.88 and -68.84 ± 6.56 kcal/mol respectively, which is determined by MM/GBSA (Fig. R4 B).

In the case of 4,9-anhydro TTX, three poses were observed during the simulation, and their binding enthalpies were -50.13 ± 9.31 , -23.48 ± 6.44 , and -35.56 ± 8.88 kcal/mol (Fig. R4 F). Notably, all these binding affinities were lower

than those of TTX. Furthermore, the simulation trajectories suggested that 4,9-anhydro TTX might not have a stable binding pose like TTX. Regarding 9-*ep*TTX, two poses were also observed, and both of them resembled the original pose, indicating that 9-*ep*TTX shared the same binding mode as TTX. However, MM/GBSA calculations revealed a significant decrease in the binding affinity of 9-*ep*TTX compared to TTX, with values of -40.02 ± 7.05 and -35.98 ± 5.05 kcal/mol (Fig. R4 D). Overall, our simulation models support the activity differences among TTX, 4,9-anhydro TTX and 9-*ep*TTX, and further emphasize the significance of the stereochemistry of 9-OH in TTX.

Figure R4: Simulation results of TTX (A, B), 9-*ep*TTX (C, D), and 4,9 anhydro TTX (E, F).

Reviewer #2: Minor comments

The following statement “Extensive pharmacological investigations, including clinical trials, have demonstrated the immense promise of TTX in pain treatment and detoxification from heroin addiction” is not fully supported by the single

reference cited. Many would argue that while TTX has some promise as a therapeutic, describing it as “immense promise” is not justified.

Our Response:

We appreciate this reviewer for his/her objective viewpoints. We conducted the revision accordingly. We also added more references of clinical reports (Ref 14,15) to support the clinical value of TTX. In addition, we changed the statement of “Extensive pharmacological investigations, including clinical trials, have demonstrated the immense promise of TTX in pain treatment and detoxification from heroin addiction” to “Extensive pharmacological investigations, including clinical trials, have demonstrated the potential promise of TTX in pain treatment and detoxification from heroin addiction”, and removed the “immense promise”.

Reviewer #3 (Remarks to the Author):

The manuscript reports a very elegant total synthesis of tetrodotoxin and 9-*epi*-tetrodotoxin. Given that the great synthetic challenges of the target molecules and their significant bio-functions, the synthesis reported will certainly be great news to natural product chemists, synthetic chemists, medicinal chemists as well as chemical biologist. Recently, Trauner has reported a concise total synthesis of tetrodotoxin (Science 377, 411-415 (2022)). From synthetic chemistry point of view, the current synthesis is , if not better, at least at the same quality as Trauner's synthesis. I am thrilled to see the flourish in the filed of TTX by the synthetic progress. I enthusiastically recommend the publish of the work in Nature Communication. It is the type of the manuscript your journal will not want to miss. Chemistry wise, the authors have done a great job. I do not have suggestion or questions. I only have some suggests regarding the presentation of the manuscript.

Our Response:

We really appreciate the positive comments and compliments from this reviewer. The recognition and appreciation of our synthesis from this reviewer are encouraging. we especially want to show our respect for his/her professionalism and responsibility.

1) Figure 1 summarized previous total synthesis of TTX. But I do not see Kishi's pioneer work. This is somehow regretful considering the leading role of Kishi in the field. I would like suggest authors to solute to Professor Kishi by adding some space to show the respect.

Our Response:

We really appreciate this reviewer for the advice of including the synthesis by Kishi in figure 1. We totally agree with this reviewer that Kishi's work and himself should be respected. We have modified the Figure 1 accordingly.

2) As another synthetic target, structure of 1a in figure 1 should also be highlighted with a light blue background.

Our Response:

Thanks for reminding us of this! We have modified this part in our revised manuscript.

3) Regarding the formation of 1b in figure 2, the arrows authors used implied a SN2 pathway, which is negotiable. In addition, the arrows depicting the formation of ortho ester is not ideal either. A simple solution is that authors just delete this structure in their final draft of the manuscript.

Our Response:

We appreciate the professional suggestions from this reviewer. We have removed the structure as suggested.

Finally, we want to express our appreciation and deep respect for all the reviewers. The overall positive assessment of our study with extremely helpful, constructive comments and suggestions have dramatically improved the scientific rigor and implications of our work. We have addressed all the concerns by providing new experimental data and by carefully considering and incorporating their constructive suggestions into the revised manuscript. We hope all reviewers will be convinced that the quality of our manuscript has been improved substantially with their help and our study now merits publication in this journal.

REVIEWER COMMENTS

Reviewer #2 (Remarks to the Author):

This is an interesting paper that describes a novel process for synthesizing TTX and possibly other heavily oxygenated polycyclic natural products. TTX is difficult to synthesize, so this process is an important technical advance. While the authors have demonstrated that the synthetic TTX is functional and the potency is at least as good, if not better, than that of commercial TTX from natural sources, the electrophysiological data is still problematic. The data from HEK 293 cells is not rigorous. As with the data presented in the previous submission, the new data shown in figure 4 (b&c) exhibits pronounced voltage-clamp errors. The data shown in panel 4d is not useful as the number of cells that express currents makes little sense. Current inhibition with TTX should not be all or none, especially for 5 and 10 nM TTX. Here one would expect a reduction in current density. The authors should conduct a concentration versus current density experiment using adequate voltage-clamp technique and estimate the IC₅₀ in a rigorous fashion.

Response letter

REVIEWER COMMENTS

Reviewer #2 (Remarks to the Author):

This is an interesting paper that describes a novel process for synthesizing TTX and possibly other heavily oxygenated polycyclic natural products. TTX is difficult to synthesize, so this process is an important technical advance. While the authors have demonstrated that the synthetic TTX is functional and the potency is at least as good, if not better, than that of commercial TTX from natural sources, the electrophysiological data is still problematic. The data from HEK 293 cells is not rigorous. As with the data presented in the previous submission, the new data shown in figure 4 (b&c) exhibits pronounced voltage-clamp errors. The data shown in panel 4d is not useful as the number of cells that express currents makes little sense. Current inhibition with TTX should not be all or none, especially for 5 and 10 nM TTX. Here one would expect a reduction in current density. The authors should conduct a concentration versus current density experiment using adequate voltage-clamp technique and estimate the IC₅₀ in a rigorous fashion.

Our Response:

We appreciate the thorough summary and professional viewpoints from this reviewer, and we are especially thankful for his/her constructive suggestions!

We agree with the reviewer's concern and conduct the experiments accordingly. To evaluate the efficiency of TTX (S) to block sodium channels, we performed a series of electrophysiological experiments.

First, we compared the blocking capability of two different sources of TTX on a single subtype of sodium channel on human HEK-Nav1.7 cells. We measured the voltage-dependent Nav1.7 currents in the absence or presence of TTX. We found that TTX (1 μ M) from different sources almost completely blocked Nav1.7 currents (Figure 4, a-f). To compare the efficiency of TTX(S) and TTX (C), we systematically measured voltage-dependent Nav1.7 currents at different concentrations of TTX (Figure 4, g-i). For the Nav1.7 currents evoked by membrane depolarization from -80 mV to +10 mV, The IC₅₀s of TTX (S) and TTX (C) were approximately 0.46 nM and 2.45 nM, respectively. These data suggest that the efficiency of TTX (S) to block sodium currents may be higher than that of TTX (C).

Figure 4, Representative traces (top) and quantitative analyses (bottom) of HEK-Nav1.7 currents under the conditions of No-TTX (a, b), 1 μ M TTX(C) (c, d), and 1 μ M TTX(S) (e, f). Inset in (c), schematic image showing the protocol of step-wise voltage-clamp from -80 mV to $+80$ mV in 10 mV increments. g-i, Quantitative analyses of concentration-dependent Nav1.7 currents evoked by membrane depolarization from -80 mV to $+10$ mV in the presence of TTX (S) (g) and TTX (C) (h). For comparison of these two curves, see (i). Note The IC₅₀s of TTX (S) and TTX (C) were approximately 0.46 nM and 2.45 nM, respectively. TTX(S) indicates synthetic TTX; TTX(C) indicates commercialized TTX from Tocris Bioscience Inc.

Second, we measured the voltage-dependent Nav1.5 currents in the absence or presence of TTX (1 μ M). We found that both TTX (C) and TTX (S) efficiently blocked voltage-dependent Nav1.5 currents (Supplementary Fig. S2).

Figure S2. The blocking efficiency between TTX(S) and TTX(C) on HEK-Nav1.5 cells. **a**, A schematic image represents an evoked step-wise protocol by voltage clamp on HEK-Nav1.5 cells from -80 mV to 80 mV in 10 -mV increments. **b**, Representative traces of HEK-Nav1.5 currents under no-TTX condition or $1 \mu\text{M}$ TTX(S) condition or $1 \mu\text{M}$ TTX(C) condition. **c**, The plot graph displays the HEK-Nav1.5 current densities evoked by series of clamped voltage from -80 mV to 80 mV under no-TTX condition or $1 \mu\text{M}$ TTX(S) condition or $1 \mu\text{M}$ TTX(C) condition. Control group, $N = 10$ cells; $1 \mu\text{M}$ TTX(S) group, $N = 11$ cells; $1 \mu\text{M}$ TTX(C) group, $N = 10$ cells. **d**, The bar graph displays quantitative comparison of HEK-Nav1.5 current densities at several clamped voltages under different TTX treatment. Control group, $N = 10$ cells; $1 \mu\text{M}$ TTX(S) group, $N = 11$ cells; $1 \mu\text{M}$ TTX(C) group, $N = 10$ cells. Error bar, Mean \pm SEM. T.Test, ** p value < 0.01 .

Finally, we examined the efficiency of TTX (S) and TTX (C) to block sodium currents evoked by voltage ramps (-70 mV to $+10$ mV) in primary hippocampal neuronal cultures. We found that our synthetic pure TTX (S) also exhibited a stronger effect in blocking the sodium currents amplitude than TTX(C) (Supplementary Fig. S3).

Fig. S3. **b**, Schematic diagram for sodium current evoked by a ramp voltage. **c**, Representative traces for sodium current amplitudes in primary cultured hippocampal neurons (DIV 14) after treatment with various TTX compounds. Black, TTX (S); Red, TTX (C). Ramp voltage from -70 mV to 10 mV over 50-ms. **d**, Quantitative analyses of sodium current amplitude in neurons treated with TTX (S) and TTX (C) with various concentrations. Cell numbers are marked on the columns. Error bars represent means \pm SEM; two-tailed unpaired t-test, *P < 0.05, **P < 0.01.

REVIEWER COMMENTS

Reviewer #2 (Remarks to the Author):

The electrophysiological experiments clearly show that the synthesized TTX can block voltage-gated sodium currents, and can block both Nav1.7 currents and sodium currents in hippocampal neurons in the nanomolar range. However, the data presented still exhibit pronounced voltage-clamp errors that compromise the interpretation of the potency of the synthetic TTX and prevent accurate comparisons of potency between commercial and synthetic TTX.

Figure S2-C is illustrative of the problem. The data presented suggests that both TTX(S) and TTX(C) substantially alter the voltage-dependence of activation. However, one can see in figure S2-B that the voltage-clamp is problematic and because of this the apparent shift in the current-voltage relationship shown in figure S2-C most likely results from the complete loss of voltage-clamp control at voltages between -40 and 0 mV.

In figure 4a the complete loss of voltage-clamp control is readily seen in the traces shown. This makes the IC₅₀ comparisons unreliable. TTX(S) may be more potent than TTX(C). The calculated IC₅₀s suggest it is 5 times more potent. These numbers cannot be trusted without adequate voltage-clamp experiments. The data shown in figure S3 also suggests that TTX(S) is more potent than TTX(C). However, here it looks like the IC₅₀ for TTX(S) is closer to 5 nM and the IC₅₀ for TTX(C) is closer to 11 nM. Here the potency of the TTX(S) is roughly ten fold lower and the difference between TTX(S) and TTX(C) is only 2 fold, not 5 fold. While these may seem relatively unimportant, with proper voltage-clamp the apparent differences between Nav1.7 currents and hippocampal currents could be clarified and the potency difference between TTX(S) and TTX(C) could be accurately measured. This seems to be important and can be easily resolved with proper electrophysiological technique.

Page 8, line 148-154. In a couple of places the term "efficiency" is used. This is not a meaningful pharmacological term. Potency would be more relevant if it is accurately measured. Efficiency can be confused with efficacy, which is not what seems to be different between TTX(S) and TTX(C).

Response letter

REVIEWER COMMENTS

Reviewer #2 (Remarks to the Author):

The electrophysiological experiments clearly show that the synthesized TTX can block voltage-gated sodium currents, and can block both Nav1.7 currents and sodium currents in hippocampal neurons in the nanomolar range. However, the data presented still exhibit pronounced voltage-clamp errors that compromise the interpretation of the potency of the synthetic TTX and prevent accurate comparisons of potency between commercial and synthetic TTX. Figure S2-C is illustrative of the problem. The data presented suggests that both TTX(S) and TTX(C) substantially alter the voltage-dependence of activation. However, one can see in figure S2-B that the voltage-clamp is problematic and because of this the apparent shift in the current-voltage relationship shown in figure S2-C most likely results from the complete loss of voltage-clamp control at voltages between -40 and 0 mV. In figure 4a the complete loss of voltage-clamp control is readily seen in the traces shown. This makes the IC50 comparisons unreliable. TTX(S) may be more potent than TTX(C). The calculated IC50s suggest it is 5 times more potent. These numbers cannot be trusted without adequate voltage-clamp experiments. The data shown in figure S3 also suggests that TTX(S) is more potent than TTX(C). However, here it looks like the IC50 for TTX(S) is closer to 5 nM and the IC50 for TTX(C) is closer to 11 nM. Here the potency of the TTX(S) is roughly ten fold lower and the difference between TTX(S) and TTX(C) is only 2 fold, not 5 fold. While these may seem relatively unimportant, with proper voltage-clamp the apparent differences between Nav1.7 currents and hippocampal currents could be clarified and the potency difference between TTX(S) and TTX(C) could be accurately measured. This seems to be important and can be easily resolved with proper electrophysiological technique.

Our response:

We thank the reviewer for pointing out the voltage-clamp errors in the original manuscript. In the revised manuscript, we have improved our voltage-clamp technique. As you can see in Figure 4a and Figure S2b of the revised manuscript (see below), the voltage-clamp was better controlled. With the improved voltage-clamp technique, we repeated our experiments by testing the effects of TTX(S) and TTX (C) on sodium currents in more cells. With these improvements in the revised manuscript, we believe it may be conclusive that the potency of TTX (S) to block Nav 1.7 and Nav 1.5 currents is still better than that of TTX (C).

Fig.4 The blocking potency between TTX(S) and TTX(C) on HEK-Nav1.7 cells. a-f, Representative traces (top) and quantitative analyses (bottom) of HEK-Nav1.7 currents under the conditions of No-TTX (a, b), 1 μ M TTX(C) (c, d), and 1 μ M TTX(S) (e, f). Inset in (c), schematic image showing the protocol of step-wise voltage-clamp from -80 mV to $+80$ mV in 10 mV increments. g-i, Quantitative analyses of concentration-dependent Nav1.7 currents evoked by membrane depolarization from -80 mV to $+10$ mV in the presence of TTX (S) (g) and TTX (C) (h). For comparison of these two curves, see (i). Note The IC₅₀s of TTX (S) and TTX (C) were approximately 2.45 nM and 3.69 nM, respectively. TTX(S) indicates synthetic TTX; TTX(C) indicates commercialized TTX from Tocris Bioscience Inc.

Fig. S2. The blocking potency between TTX(S) and TTX(C) on HEK-Nav1.5 cells. **a**, A schematic image represents an evoked step-wise protocol by voltage clamp on HEK-Nav1.5 cells from -80 mV to 80 mV in 10 -mV increments. **b**, Representative traces of HEK-Nav1.5 currents under no-TTX condition or 1 μ M TTX(S) condition or 1 μ M TTX(C) condition. **c**, Plot graph displays the HEK-Nav1.5 current densities evoked by series of clamped voltage from -80 mV to 80 mV under no-TTX condition or 1 μ M TTX(S) condition or 1 μ M TTX(C) condition. **d**, Bar graph displays quantitative comparison of HEK-Nav1.5 current densities at several clamped voltages under different TTX treatment. Error bar, Mean \pm SEM. T.Test, ** p value < 0.01 .

Page 8, line 148-154. In a couple of places the term "efficiency" is used. This is not a meaningful pharmacological term. Potency would be more relevant if it is accurately measured. Efficiency can be confused with efficacy, which is not what seems to be different between TTX(S) and TTX(C).

Our response:

We thank the reviewer. We agree with the reviewer that "efficiency" may not be a meaningful pharmacological term. In the revised manuscript, we replaced "efficiency" with "potency", which has been highlighted in yellow.

REVIEWERS' COMMENTS

Reviewer #2 (Remarks to the Author):

The revised figure 4 is improved, although there is still some evidence of voltage-clamp problems, but the data are acceptable. However, while the new data does indicate that the synthetic TTX is more potent than the commercial batch, the figure would benefit from using a log curve for the a-axis (TTX concentration) for the data shown in g, h and i, as is standard practice. It is also important to state if the curves are significantly different.